# Rikkosan’s Short-Term Analgesic Effect on Burning Mouth Syndrome: A Single-Arm Cohort Study

**DOI:** 10.3390/biomedicines12051013

**Published:** 2024-05-04

**Authors:** Tatsuki Itagaki, Keisuke Nakamura, Tougo Tanabe, Takumi Shimura, Yu Nakai, Ken-ichiro Sakata, Jun Sato, Yoshimasa Kitagawa

**Affiliations:** Department of Oral Diagnosis and Medicine, Faculty of Dental Medicine, Hokkaido University, Kita-13 Nishi-7, Kita-ku, Sapporo 060-8586, Japan; titagaki@den.hokudai.ac.jp (T.I.); knakamura@den.hokudai.ac.jp (K.N.); tanabet@den.hokudai.ac.jp (T.T.);

**Keywords:** burning mouth syndrome, nociplastic pain, pharmacotherapy, Kampo medicine

## Abstract

Burning mouth syndrome (BMS) is a chronic oral pain disorder. There is a theory that BMS is a form of nociplastic pain. A standard treatment for BMS has not yet been established. Kampo medicine is a traditional oriental medicine. The purpose of this study is to evaluate the effectiveness of Rikkosan—a traditional Japanese herbal medicine (Kampo)—in the treatment of BMS. A single-center retrospective study was conducted on 20 patients who were diagnosed with BMS and treated with Rikkosan alone (total daily dose; 7.5 g) three times daily for approximately 4 weeks (29.5 ± 6.5 days). Rikkosan was dissolved in hot water and taken internally. They had an average age of 63 years, and 90% were being treated for other illnesses, but their medication status was the same during this study period, except for Rikkosan. No adverse events were observed in patients. Numerical rating scale (NRS) or visual analog scale (VAS)/10 scores decreased significantly between the time of the initiation of Rikkosan and one month after (−2.1 ± 1.2, *p* < 0.05). Rikkosan has a short-term effect of reducing NRS by two levels in BMS patients.

## 1. Introduction

Burning mouth syndrome (BMS) is defined in the International Classification of Orofacial Pain, 2020, as “idiopathic orofacial pain with intraoral burning or dysesthesia recurring daily for more than 2 h per day and more than 3 months, without any identifiable causative lesions, with or without somatosensory changes” [1,2]. The development of research diagnostic criteria for BMS was published in 2021 [3]. Previously, it was divided into primary and secondary BMS [3]. Intraoral burning is a result of a range of underlying causes of lesions, such as candida infection, oral lichen planus, hyposalivation, contact mucosal reactivity, medications, anemia, deficiencies in vitamin B12 or folic acid, Sjögren’s syndrome, diabetes, and hypothyroidism [3]. However, the correct diagnosis of BMS is a diagnosis of exclusion, and only primary cases are diagnosed as true BMS. 

There are three pain classifications for chronic pain [4]. Nociceptive pain often involves obvious damage, which is easy to see on the body surface. BMS does not show obvious damage, so it is unlikely to be nociceptive pain, but the possibility of nociplastic pain that develops afterward cannot be ruled out. There is a theory that BMS is neuropathic pain [5]. BMS is probably nociplastic pain or neuropathic pain. There is no established treatment for nociplastic pain. However, therapeutic drugs for neuropathic pain are often used to treat nociplastic pain.

There are various reports on effective ways to treat BMS patients [6,7,8,9,10,11]. One of the treatment options for BMS is the topical and systemic application of clonazepam [6,7,8,9,10,11]. There are a variety of options available such as systemic selective serotonin reuptake inhibitors (SSRIs), zinc replacement therapy, alpha-lipoic acid, aloe vera, hormone replacement therapy, cognitive behavior therapy, and acupuncture, as well as other treatment options [7,8]. However, some suggest that we should not use clonazepam and alpha-lipoid acid for people with burning mouth syndrome [10,11,12]. Y Cui et al. conducted a meta-analysis of clonazepam, but the heterogeneity was large [11], making it undesirable to pool the results. Therefore, the current treatment management is debatable [9,10,11,12]. There is the fact that the management of BMS is still not efficacious with the traditional management options [8,9,10]. There is still no treatment for BMS whose usage has been confirmed in phase 2 trials. There are many studies based on the amount of change in the visual analog scale (VAS) [9,10,11]. However, there is no significant evidence reporting on the effectiveness [9,10,11].

Rikkosan is a traditional Japanese (Kampo) medicine used to control oral pain [13,14,15]. Rikkosan has anti-inflammatory effects, but the exact mechanism is still unknown [11,12,13]. The surface anesthetic action of saishin, one of the components of Rikkosan, may reduce the pain [13,14,15]. In addition, Rikkosan contains two ingredients with analgesic effects (ryutan (glycyrrhiza) and kanzou (Japanese gentian)) [14]. The sedative action of shoma and kanzou might reduce pain [15].

One of our previous studies evaluated the change in pain intensity using the numerical rating scale (NRS) [13]. After using Rikkosan for one month, the average change in NRS was an approximately two-point reduction in pain intensity [13]. Other previous studies evaluated the improvement rate defined as reduced subjective VAS to <50% of baseline before Rikkosan treatment [14]. The sample sizes for each study were 32 and 48 people [13,14]. The results of these studies suggested that Rikkosan is a candidate drug for the treatment of nociplastic pain or BMS.

## 2. Materials and Methods

### 2.1. Patients Selection

Patients were examined and diagnosed in our department according to the criteria of The International Classification of Orofacial Pain, 1st edition (ICOP) [2,3]. Patients were considered to have secondary BMS if systemic and psychosocial factors associated with secondary BMS and structural disorders in the oral cavity were found during examination. Patients with residual symptoms after antifungal therapy and replacement therapy for deficiency factors such as trace metals and vitamin B12 were diagnosed as BMS when blood tests were normal. 

The patients visited the Department of Oral Medicine, Hokkaido University Hospital, between August 2019 and March 2023. Patients with an underlying medical condition, based on an interview, or any abnormalities found through our diagnostic algorithm were diagnosed with secondary BMS. The classification of secondary BMS was based on the proposal of Currie et al. [3]. Twenty female primary BMS patients who had clear medical records of their treatment were enrolled in this study. The patients were not taking any other medications for BMS. In Japan, oral medicine has been established as a subspecialty of oral surgery. All of them were diagnosed at the second or third visit by multiple oral surgeon specialists with over 10 years of experience and accredited by the Japanese Society of Oral and Maxillofacial Surgeons (S.K.-i. and J.S.).

### 2.2. Procedure

The patients were diagnosed with primary BMS and started treatment with single-agent Rikkosan (2.5 g of Rikkosan [Tsumura, Tokyo, Japan]) three times daily (total daily dose, 7.5 g/day). Rikkosan was dissolved in hot water and taken internally. After drug therapy, the patients’ compliance with taking the medication was confirmed. There were no changes in the medication status of other drugs during treatment with Rikkosan.

### 2.3. Study Variables

Various factors, such as patient characteristics (age, sex) and clinical parameters (dosing period, treatment outcome, and side effects), were retrospectively examined.

The patients were 43–80 (63.3 ± 13.4) years old, and the median duration of pain complaints was 14 months (range: 3 months to more than 10 years). A total of 90% of them were undertaking other medications for other diseases. The medication status was the same during this study period. In other words, Rikkosan was just added to the medication. They were started on approximately 4 weeks (29.5 ± 6.5 days) of treatment with Rikkosan.

The effectiveness of the treatment was assessed by referring to changes in NRS or VAS/10 scores. NRS or VAS/10 scores were evaluated by asking patients to assess the degree of pain they were currently experiencing, with 0 being no pain and 10 being the worst possible pain. NRS or VAS/10 scores were measured at starting the treatment with Rikkosan and at 1 month after.

### 2.4. Statistical Analysis

Statistical analyses were performed using Excel (Microsoft^®^ Excel^®^ for Microsoft 365MSO (version 2306 build 16.0.16529.20164, 64 bit)) and R version 4.0.3 (2020-10-10) (Copyright © 2020, The R Foundation for Statistical Computing). The effectiveness of the pain evaluation index was evaluated based on the presence or absence of ceiling and floor effects. The ceiling effect was determined when the mean plus exceeded the maximum value of the measurement, and the floor effect was determined when the mean minus one standard deviation (SD) exceeded the minimum value of the measurement. Based on our previous exploratory research [13], the average effect of Rikkosan was estimated to be a two-level reduction in NRS, and the sample size was designed with a standard deviation of 2.6, an alpha error of 5%, and a power of 90%. The calculated sample size was 20 people. The 20 selected patients were the 20 patients with the lowest patient identification number at our hospital who met this study criteria. A paired t test was performed to assess the hypothesis on the change in mean NRS scores at approximately a one-month interval.

Null hypothesis: The reduction in pain intensity is 2 points.Alternative hypothesis: The reduction in pain intensity is 2 points or more.

As a secondary analysis, changes in pain intensity were evaluated using Kendall’s coefficient of concordance, and Friedman’s test was performed to compare the center of grade distribution before and after administration.

### 2.5. IRB Approval and Ethics

This retrospective study was conducted with the approval of the Hokkaido University Hospital Independent Clinical Research Review Committee (Approval No. 023-0331). All the study procedures were performed in accordance with the principles of the Declaration of Helsinki.

## 3. Results

As summarized in Table 1, the ceiling effect and the floor effect were not observed. Therefore, the assessment of pain worked well. Figure 1 showed changes in NRS or VAS/10 scores between the time of the beginning of Rikkosan and one month after. 

Table 2 summarizes the changes in pain scores for the average treatment effect with Rikkosan in 4 weeks and a qualitative assessment of the change. A significant difference was observed in pain evaluation before and after treatment (*p* < 0.05). In other words, inferential statistics ensured reproducibility of pain intensity reductions of two points or more. Kendall’s coefficient of concordance was 0.85. As summarized in Table 2, an 85% improvement rate was achievable (*p* = 3.7 × 10^−5^) if a slight improvement was set as a cutoff. However, if the cutoff was an improvement of two or more, it was 50% in this study.

## 4. Discussion

This is the first study to assess the working hypothesis that Rikkosan leads to a two-point reduction in pain in patients with BMS. Rikkosan treatment showed a two-level reduction in NRS or VAS/10 scores for BMS. This result indicated that Rikkosan could be the first good therapeutic option for BMS in the short term. Our results statistically showed that Rikkosan produced a two-point reduction in pain intensity. The response rate may be used to indicate the therapeutic effect. However, the response rate is an ambiguous result that is binarized by the cutoff value. The specific effect size is unknown. When the improvement rate is used as an evaluation index, there is a problem with the cutoff value. On the other hand, this study performed quantitative evaluation using NRS or VAS. Consequently, the average treatment effect could be predicted. Due to the study design, 90% reproducibility is expected under similar conditions. The statistics regarding the change in NRS were consistent with the results of our previous study [13].

Traditional Chinese medicine developed the use of medicinal plants [16]. Traditional Japanese medicine, also called Kampo medicine, derives from traditional Chinese medicine [16]. In recent years, Kampo medicine has been attracting attention in the West as well [16]. Rikkosan contains licorice. Licorice contains glycyrrhizic acid, which is hydrolyzed to glycyrrhetinic acid (GRA) in the intestine [16]. Therefore, a side effect of Rikkosan is pseudoaldosteronism, so regular blood tests are required. The electropharmacological profile of licorice can be explained by Na^+^, Ca^2+,^ and K^+^ channel blockade. Ion channels exist in pain receptors [17]. These ion channels may be one of the targets of Rikkosan.

This study has several limitations. An important limitation was the short-term follow-up. The general flow of research is shown. First, we considered the details of the case. Next, we generated hypotheses using descriptive statistics. This was also the case in our previous study [13]. We used the content of the phase 2 trial. Based on the generated hypothesis, a null hypothesis and an alternative hypothesis were set, and inferential statistics calculations were performed. The purpose was to estimate the characteristics of an infinite population. In this phase 3 trial, many examples showed significance compared to the placebo. Single-arm cohort studies are prone to false positives. However, this study did not include a placebo group. The before–after comparison became significant, but there may be no superiority over the placebo. To control the probability of false positives and false negatives and to detect meaningful differences, the study preset parameters before data collection. The minimum clinically meaningful change in pain intensity was set at two points. A minimum three-point NRS reduction is usually considered clinically meaningful in the pain field. A three-point NRS reduction was statistically a large difference, and the difference can be detected with a small number of cases tested. Based on the simulation results, the central limit theorem was established in this study with 20 people. If the sample size is larger than the preset sample size, false positives will occur due to overdetection. The study design had a limit of two-point NRS reduction. Therefore, the purpose of this study was to confirm the hypothesis that Rikkosan leads to a two-point decrease in pain intensity. The working hypothesis was based on our previous preliminary research [13]. In other words, the study removed as much bias as possible. By proving this working hypothesis, the effects of Rikkosan can be examined. This study increases external validity and allows consideration of Rikkosan as a treatment for BMS.

Rikkosan has a bitter taste, so some people do not like it. Taste preferences may be used to determine responders. Some patients did not want further treatment, but others still required treatment. The duration of treatment effects may vary depending on the patient. Therefore, further research on additional treatments is needed. The effect of Rikkosan can be calculated by subtracting the effect of the placebo from this result. The crossover design of placebo and Rikkosan allows for direct calculation of the effect of Rikkosan. The effect of the placebo is unknown in this study. It would be more constructive to verify the placebo effect in the next study. However, careful consideration should be given to evaluating the quantitative effect of Rikkosan before conducting the next trial.

## 5. Conclusions

This study quantitatively evaluated the effects of Rikkosan, including some bias. The results of our study suggested with statistical reproducibility that Rikkosan can reduce pain intensity in patients with BMS by two levels.

## Figures and Tables

**Figure 1 biomedicines-12-01013-f001:**
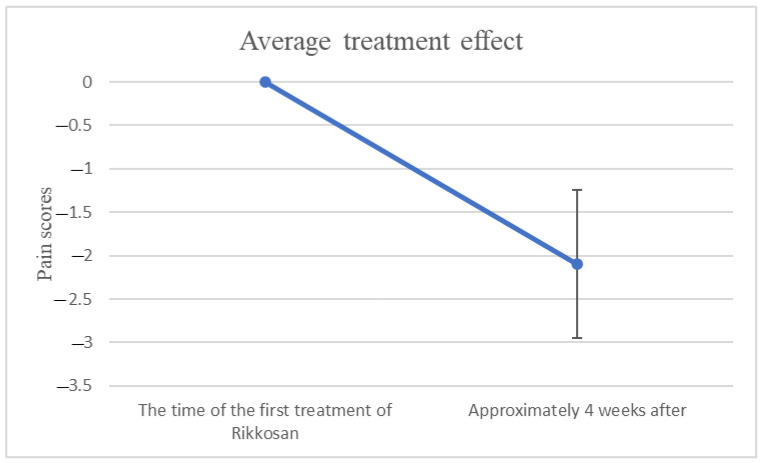
This showed the treatment results. An average improvement of two levels of NRS is expected (*p* < 0.05). The error bars showed a 95% confidence interval.

**Table 1 biomedicines-12-01013-t001:** Descriptive statistics for pain scores.

Pain Scores	Pre	Post
Mean (SD)	5.9 (2.5)	3.8 (2.2)

**Table 2 biomedicines-12-01013-t002:** Verification results regarding pain scores.

Variable	Values
Paired *t* test’s *p*-value	5.5 × 10^−5^
The difference	2.1
Effect size	1.2
Standard deviation of change	1.8
Standard error of change	0.41
Kendall’s coefficient of concordance	0.85
Friedman’s test’s *p*-value	3.7 × 10^−5^

## Data Availability

The data presented in this study are openly available in the Appendix A.

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
