# Peer review of "Rikkosan’s Short-Term Analgesic Effect on Burning Mouth Syndrome: A Single-Arm Cohort Study"

_biomedicines, 2024, doi:10.3390/biomedicines12051013_

Round 1
Reviewer 1 Report
Comments and Suggestions for Authors
No comments.
Author Response
Dear Reviewer 1,
Thank you for giving us the opportunity to strengthen our manuscript with your valuable comments. We have worked hard to incorporate your feedback and hope that these revisions persuade you to accept our submission.
Please understand our study.
This research is not about discovering new things. This is a hypothesis verification type study.
This study found the reduction in pain intensity is 2 points or more by Rikkosan.
H0: The reduction in pain intensity is 2 points.
H1: The reduction in pain intensity is 2 points or more.
The power was sufficient. Our study accepted alternative hypothesis (H1). The reason for this is that the p value was less than 5%. If the p value was 5% or more, the negation of alternative hypothesis (The reduction in pain intensity is less than 2 points) is accepted.
Reviewer 2 Report
Comments and Suggestions for Authors
This study has several methodological flaws (see below). Moreover, it is not expressed in a scientifically valid language.
Abstract:
- this sentence (treated with rikkosan alone (7.5 g) three times daily for approximately 4 weeks. They were treated with rikkosan alone approximately 4 weeks (29.5 ± 6.5 days) for the initial treatment) is repeated twice.
Introduction:
- These sentences are not expressed in a correct scientific English language (Tests have false positives and false negatives. Even if the algorithm is followed, misclassifica- tion bias is inevitable. In our hospital, it is common to diagnose BMS by excluding other 35 diseases whenever possible.) I suggest omitting them.
- Line 38: what does it mean “For another treatment”?
- This sentence (However, some suggest that cautious use of clonazepam and alpha-lipoid acid be 42 used in the treatment of burning mouth syndrome”) is not expressed in correct English language
- These two sentences (“There are various reports on effective ways to treat BMS patients [4–9].”and “ There is the 45 fact that the management of BMS is still not efficacious with the traditional management 46 options [8–10] “ contradict each other).
- “There are many studies on quantitative assessment, based on the amount 48 of change in visual analogue scale (VAS), but there are no reports of high evidence for the 49 effectiveness [7–9].” Not in English language
- What did this study find? “The other our previous studies evaluated qualitatively, which evaluate the improvement rate defined as reduced subjective VAS to <50% of baseline before rikkosan 55 treatment [12].” Both of them are quantitative assessment.
- Before introducing the study, Rikkosan needs to be introduced with a logic flow.
- The passage “However, if there are 65 fewer than 20 people, the sampling becomes more biased, increasing the bias. The study 66 design had a limit of 2-point NRS reduction. Therefore, the purpose of this study was to 67 confirm the hypothesis that rikkosan leads to a 2-point decrease in pain intensity. The 68 working hypothesis was based on our previous preliminary research” needs to be totally changed. First of all, what is the aim of the study? The aim is not to confirm a hypothesis. The aim is one thing; then the hypothesis is indicated. If the authors already know that N = 20 is not enough, why didn’t they increase the sampel size? They are basically saying here that N=20 is biased. This totally discredits their study and their findings.
Methods:
- “Patients were examined and diagnosed in our department according to the criteria 72 of The International Classification of Orofacial Pain, 1st edition (ICOP) [2,3].” Diagnosed with what condition?
- This sentence “At our facility, multiple oral medicine and oral surgeons specialists with over 79 10 years of experience and accredited by the Japanese Society of Oral and Maxillofacial 80 Surgeons make final diagnoses and provide treatment.” Speaks to the lack of internal validity, as several providers have provided the diagnosis.
- This passage from lines 79 to lines 89 basically repeats the same concept as above.
- It is not clear why the authors are mentioning secondary BMS, if they only included primary BMS
- Line 91-94: authors need to state that the patients were not taking any other medications for BMS
- Lines 98 to 101 need to be in the results, not in the methods
- What is this? “to compare the center of grade distribution before 123 and after administration.”
- The ceiling effect needs to be explained in the outcome, if this was an outcome assessed.
Results:
Include table 2 results in table 1, otherwise the p values do not make any sense. Also, transform into <.0001. How was the effect size calculated?
Discussion:
- This sentence is not true (“This is the first study to assess the working hypothesis which rikkosan leads to a 2- 146 point reduction of pain in patients with BMS.”), as other studies have been reported in the introduction on this treatment.
- This sentence is also not true (“This result indicated that rikkosan could be a 148 first good therapeutic option for BMS in the short term.”), considering that the authors have declared that a 3 point change on a 0-10 NRS is considered significant.
- This sentence “However, this does not depend on how effective they 151 are individually, but rather they are the same if they are effective above a certain level.” Is not expressed in correct English language
- This passage (from line 158-167) does not fit here. By the way, the fact that BMS is neuropathic in nature is not agreed by everyone in the literature.
Comments on the Quality of English Language
Proposed in my comments
Reviewer 3 Report
Comments and Suggestions for Authors
The MS, "Rikkosan's Short-term Analgesic Effect of Burning Mouth Syndrome: A single-arm cohort study," has a great purpose: to report Rikkosan as an effective treatment for BMS, a disease still unknown and without a standard therapeutic protocol. The authors conducted an interesting retrospective study on 20 patients with primary BMS. Their work is valuable and helpful for BMS patients. However, this MS needs a suitable data presentation in all sections and requires extensive revision.
In a case study, the reader expects to find a dynamic study with well-established inclusion criteria, justifications, and a clear presentation of the results before and after. In addition, the suitable observations are supported by relevant data analysis (descriptive and correlations).
The present work cites 3 previously published articles (references 11-13) on the same subject, which present all data excellently and clearly. Maybe to avoid similarities with the previously published studies, the present work partially shows the data regarding the cohort of 20 BMS patients, continuously citing them. Thus, the reader can understand the presented data only after reading the previous articles.
The abstract is blurred, and the data presentation is unclear. The name of traditional Japanese medicine is edited with capital letters, while the name of the traditional drug combination is in lowercase; thus, the reader has difficulties deciphering which herbal drug is Kampo or rikkosan. Moreover, it is unclear if 7.5 g is a one-time dose, repeated three times a day, or the total daily dose. The way of rikkosan administration is not shown; the evaluation mode also has a dangling presentation, and the reader must access the previous articles to find these details. All data are presented as a mean +/- SD, including the treatment duration and result.
Introduction
After a background regarding BMS updates (less extensive than in the current version), the authors are encouraged to briefly present Kampo medicine and the most used remedies. If possible, they can provide details on the rikkosan composition, pharmacological properties, and benefits. The previously published studies (lines 51-56) could be described in the Discussion, not the Introduction. The same comment is available for lines 63-69.
Material and methods
50% of the subsection Patient selection is subject to Discussion (lines 72-70).
The following data did not offer suitable patient selection information- they are unclear and superficial; the most important data are missing.
Lines 91-94: the rikkosan dose is constant; thus, avoid the term "initiation," more available for variable doses in the therapy period. Moreover, the administration method and the pharmaceutical formulation of rikkosan were requested.
Lines 101-102: please clarify: does another round of treatment exist? If it is not discussed in the present study, please correct it.
Statistical analysis only shows the mean values and all statistical coefficients, which are commonly stated in the Supplementary Material. In the present MS, the situation is vice versa: the individual results (before/after) are in the Supplementary Material.
Results:
The essential data are missing, and the Results section occupies only 1 page from the whole MS.
The authors could present the obtained individual results before/after using a graphical representation because the reader expects them.
Moreover, they must comment on displayed individual data, making global correlations between age, other drug consumption, pain duration, and the pain score (and difference) after 1 month of treatment with triclosan.
Then, they could analyze the mean values per group and show the median pain reduction with 2 levels.
Discussions and Conclusions should be reformulated after MS revision. In the current version, the lack of results does not support the conclusions.
Lines 32-33, 42-43, 51, 54, 119: Please reformulate, avoiding repetition, to clarify the statements.
Moreover, the whole MS needs a rigorous revision to remove all repetitions and misprints. For example, (Kampo) is repeated in the MS text: lines 12 (Kampo) 57 (kampo).
A term can be explained once in the text, at its first appearance (lines 57 and 169).
Lines 12, 57, 168, 169: Kampo/kampo; please maintain the text uniformity.
Reviewer 4 Report
Comments and Suggestions for Authors
The communication manuscript by Itagaki T. et al., entitled: Rikkosan’s Short-Term Analgesic Effect of Burning Mouth Syndrome: A Single-Arm Cohort Study, presents the results of daily administration of rikkosan for 4 weeks to patients who have been diagnosed with BMS.
The study is interesting and useful for the improvement of the chronic oral pain disorder.
Still, some issues must be addressed. First of all, English language is not very clear in all manuscript. For example, the following sentences must be rephrased, as they are not clear:
However, some suggest that cautious use of clonazepam and alpha-lipoid acid be used in the treatment of burning mouth syndrome
One of our previous studies evaluated quantitatively, which evaluate the change of the pain intensity using numerical rating scale (NRS)
The other our previous studies evaluated qualitatively, which evaluate the im-provement rate defined as reduced subjective VAS to <50% of baseline before rikkosan treatment.
The abstract contains a repetition which seams senseless: A single-center retrospective study was conducted in 20 patients who were diagnosed with BMS and treated with rikkosan alone (7.5 g) three times daily for approximately 4 weeks. They were treated with rikkosan alone approximately 4 weeks (29.5 ± 6.5 days) for the initial treatment.
It must be mentioned the pharmaceutical dosage form and the way of administration of rikkosan. Was it topically applied or orally swallowed or injected? What kind of product was used and where it was prepared?
What exactly means 100 in the following sentence: NRS or VAS/10 scores were evaluated by asking patients to assess the degree of pain they were currently experiencing, with 0 being no pain and 10 or 100 being the worst possible pain.
Comments on the Quality of English LanguageEnglish language needs moderate editing
Round 2
Reviewer 2 Report
Comments and Suggestions for Authors
The authors have provided a rebuttal of their manuscript. I still have some concerns with the manuscript.
LIne 57: this is not correct. There are plenty studies where gabapentin (a neuropathic med) was tested, for example. TO just mention one neuropathic med.
Line 54: the reason why the theory of BMS not being a neuropathic condition is unreliable is not because in that study there was a difference in age group.
Line 53: only two theories are discussed: neuropathic and nociplastic. It does not even make any sense to introduce the theories after the paragraph on Rikkosan.
Comments on the Quality of English LanguagePublishing and editing office needs to carefully review and improve the language of the paper.
Reviewer 3 Report
Comments and Suggestions for Authors
The reviewer appreciates the authors' efforts to respond to the previous comments. However, they can still make some improvements by reading the previous review report attentively.
Reviewer 4 Report
Comments and Suggestions for Authors
Thank you for accordingly revising the manuscript
Comments on the Quality of English LanguageThe English language still needs moderate editing
Author Response
Dear Reviewer 4,
Your comments are helpful. Your suggestions have improved our manuscript. The English may be better in the final edit.
Again, thank you for giving us the opportunity to strengthen our manuscript with your valuable comments. We have worked hard to incorporate your feedback and hope that these revisions persuade you to accept our submission.